# A refined approach for evaluating small datasets via binary classification using machine learning

Steffen Steinert [1,2]*, Verena Ruf [1], David Dzsotjan [1], Nicolas Großmann [3], Albrecht Schmidt [4], Jochen Kuhn [1], Stefan Küchemann [1]

1 Chair of Physics Education, Ludwig-Maximilians-Universität München (LMU Munich), Munich, Germany, 2 Department of Electrical and Computer Engineering, RPTU Kaiserslautern-Landau, Germany, 3 Smart Data & Knowledge Services, German Research Center for Artificial Intelligence, Kaiserslautern, Germany, 4 Human-Centered Ubiquitous Media, Ludwig-Maximilians-Universität München (LMU Munich), Munich, Germany

* steinert.steffen@physik.uni-muenchen.de

**Data Availability Statement:** The Python implementation of the methodology developed and used in the study as well as the code to produce the results, is available under https://osf.io/8rkjb/.

## Abstract

Classical statistical analysis of data can be complemented or replaced with data analysis based on machine learning. However, in certain disciplines, such as education research, studies are frequently limited to small datasets, which raises several questions regarding biases and coincidentally positive results. In this study, we present a refined approach for evaluating the performance of a binary classification based on machine learning for small datasets. The approach includes a non-parametric permutation test as a method to quantify the probability of the results generalising to new data. Furthermore, we found that a repeated nested cross-validation is almost free of biases and yields reliable results that are only slightly dependent on chance. Considering the advantages of several evaluation metrics, we suggest a combination of more than one metric to train and evaluate machine learning classifiers. In the specific case that both classes are equally important, the Matthews correlation coefficient exhibits the lowest bias and chance for coincidentally good results. The results indicate that it is essential to avoid several biases when analysing small datasets using machine learning.

## Introduction

In recent decades, machine learning (ML) has become one of the most widely used tools for data analysis, as it significantly broadens classical statistical methods for analysing large datasets [1]. This is evident in various fields such as protein research [2], where ML models are used to analyse protein properties, and in education [3], where ML aids in understanding student performance and learning patterns. However, the analysis of small datasets with ML is not straightforward and can be subject to biases and methodological errors. This may lead to results that are not generalizable to other datasets and, therefore, are of limited use [4, 5].

Various decisions are relevant when analysing small datasets with ML. Apart from choosing an algorithm, it is important to consider the evaluation metrics and the evaluation method to

**Funding:** The author(s) received no specific funding for this work;.

**Competing interests:** The authors have declared that no competing interests exist.

avoid overly optimistic results. The variance of the data can be used in this step to identify good methods to avoid such misjudgements. Furthermore, methods from classical statistics, such as hypothesis testing with $p$-values, can indicate whether the results are random or statistically unlikely.

This paper aims to present a general approach and is therefore not specific to one algorithm but rather focuses on the general issues that stem from the evaluation methodology for binary classification. Exemplary algorithms for binary classification include random forests and Support Vector Machines (SVM). While different binary classifiers operate differently, they all share common challenges that arise from the evaluation method, such as varying evaluation results depending on the random seed [6]. To illustrate these issues, we have chosen to focus on SVMs, because they generally yield high scores for various evaluation metrics used in this paper. Apart from that, no broader generalisations can be made about the superiority of an SVM over a random forest, as their performance may vary depending on the dataset [7].

Another aspect to consider when evaluating data using ML is the choice of evaluation metric the classifier should be trained on. The performance metrics differ in terms of their sensitivity to different features of the data. For example, accuracy (ACC) does not take into account imbalances in the dataset: for highly imbalanced datasets, one might obtain a very high ACC even though the algorithm trivially predicts only the same class [8]. There are many advantages and disadvantages of common metrics, such as ACC, that are summarised in the Supporting information.

Apart from the evaluation metrics, it is important to consider the evaluation method. Each evaluation method has different properties, therefore, they are not equally suitable for various types of datasets, and some methods may produce highly variable or biased results [5]. For example, the train/test split method has a large variance for small datasets [6] and, therefore, does not provide a precise estimate of classifier performance when analysing small datasets with ML.

Typically, the classification error decreases as the number of data points increases [9], but a strong negative correlation between the number of data points and high reported ACC was found in an evaluation of the results of several studies on autism [5]. This correlation is rather unexpected for ML algorithms and might be explained by an underlying bias [5] caused by commonly employed methods like hyperparameter tuning or feature selection which can lead to overestimate classifier performance if not used correctly. This negative correlation indicates that misuse of feature selection and/or hyperparameter tuning might be common. However, there are ways to avoid too optimistic evaluations, such as nested cross-validation (nCV) [10, 11]. In this method, an inner cross-validation loop is used to estimate the model performance for each hyperparameter combination, and an outer cross-validation loop is used to estimate the generalisation performance of the selected hyperparameters. However, for small datasets, this method is susceptible to variance and repeating this method several times with the same data but different random seeds yields different results. Therefore, it necessary to repeat this method with different cross-validation (CV) splits [12]. Even for uncorrelated features and target variables, the data structure can lead to higher results than one would expect from random data [13]. In particular, if the value of the metric produced by the classifier is close to what would be expected from a random prediction, there is a chance that the performance of the classifier will not generalise to new data. Standard methods such as train/test splitting or nCV do not provide information on the probability that the trained algorithm will randomly perform well in the validation process [14]. During the evaluation of the classifier performance, there may be inaccuracies that lead to an overly optimistic estimation of the prediction quality, which can have crucial social and medical consequences, such as those related to the diagnosis of COVID-19 from medical images [4].

In classical statistics, one of the most commonly used methods for hypothesis testing is the determination of the *p*-value [15]. The *p*-value indicates the probability that a null hypothesis —for example, that a treatment in an intervention group has no effect compared to a control group—is true. Below a certain threshold, such as 5%, the null hypothesis is rejected.

In ML, the determination of a probability comparable to the *p*-value is not well established. However, such a probability can contain valuable information regarding the likelihood of obtaining randomly high scores which do not generalise. Although similar evaluation methods exist [13], they are usually not applied in practice [14]. Particularly for small datasets, which are common in education research, an estimated good prediction quality might also be due to chance [14]. This can happen if the features and the target variables are independent—that is, there is no difference between classes [13]—or if the classifier was not able to exploit the dependency in the data [16]. It is important to quantify a probability for this null hypothesis to be true, because in this case, the results obtained from a high probability dataset would likely not generalise to new data.

Currently some authors evaluate their work using only simple train/test splits or CVs even when working with small data and obtaining results close to the value of random prediction (e.g. [17]) or comparing results differing only slightly (e.g. [18]). Previous works showed that different methods exist to avoid pitfalls [5], to reduce the variance of results [12] and to quantify a probability that the results do not generalise [13].

In this work, we combine the results of several previous studies (e.g. [5, 12, 19–23]) with our own experimental results on both synthetically generated and real-world datasets to demonstrate crucial factors in ML-based classification of small datasets. We demonstrate that individual methods on their own (e.g. [5, 12, 13]) are not always sufficient for a generalizable ML-based classification of small data sets. Instead, we show that it is necessary to unify all these methods in form of a repeated nested cross-validation (rnCV) and use a non-parametric permutation test [13, 24], analogous to the *p*-value in classical statistics. In this way, we intend to raise awareness of these problems avoided by this method.

## Materials and methods

### Evaluated data

Several synthetically generated datasets as well as real-world datasets are examined. Predictable synthetic datasets are generated to investigate the dependence of the prediction quality and the probability of the null hypothesis on the metric and the evaluation method used (see Fig 1). These balanced binary datasets are constructed with distinct feature distributions per class. Initially, half the dataset forms class 1, where the features follow a Gaussian distribution (mean = 0, stddev = 1). Subsequently, class 2 is generated, mirroring class 1's generation process but altering the mean to 0.2. Post-generation, data points undergo random shuffling. To find the dependence of the score on the number of data points, we created several datasets containing 10 features with the following sizes: 25, 30, 40, 45, 50, 90, 130, 170, 210, 250, 290 and 330 data points. We choose this variable step size to find a compromise between computation time and resolution. In addition, to determine the mean and variance of the scores between the different datasets for different numbers of data points and features, 100 datasets of each dimensionality are generated and analysed. Of course, it would be advisable to analyse more datasets to obtain a more stable mean of the scores, but a balance has to be established between a sufficient number of datasets and the computational time required.

The 'Conceptual understanding of Electromagnetism Supported by Augmented Reality experiments' (CESAR) [25], 'Breast Cancer Wisconsin (Diagnostic)' (BCWD) [26] and

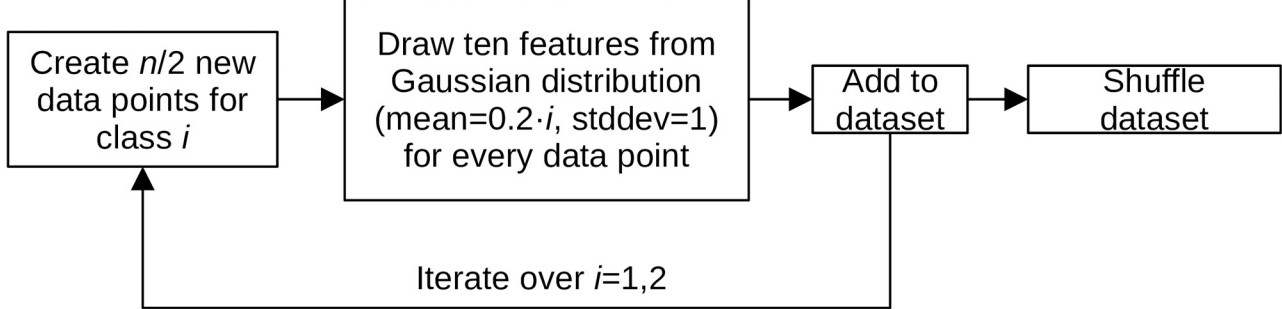

**Fig 1. Synthetic dataset creation process.** In a first step, half of the target dataset size 'n' is created for the first class. For each data point, 10 features are then added whose values are drawn from a Gaussian distribution with a mean of 0 and a standard deviation of 1. After adding these points to the dataset, half the target dataset size data points are created for class 2, and for each point, 10 features are drawn from a Gaussian distribution with a mean of 0.2 and a standard deviation of 1. These points are also added and the resulting dataset is shuffled.

'Modified National Institute of Standards and Technology' (MNIST) [27] datasets are used to demonstrate the evaluation method on real datasets and to investigate the dependence of the results on the metric in practice. The CESAR dataset is a multi-part educational science dataset containing, among other things, the results of a test of representational ability on vector field plots and field lines [28], the results of a concept test of electromagnetism, as well as demographic data of the students. This test was administered online using the Unipark survey software [29]. A total of 515 students from the University of Saarland, the University of Kaiserslautern and the Swiss Federal Institute of Technology Zurich participated in the study and are included in the dataset. For the ML, only the 12 questions of the representational competence test, the completion time, and the self-assessment of the data quality were used. The column names of the features used are given in parentheses: duration (duration), data quality (DQ) and answers to questions 1–12 (repko_1–repko_12) with questions 4 and 5 split into four (4a–4d) and three (5a–5c) subquestions. In this paper, the responses to question 5b are predicted using the responses to the remaining items of the representational competency test, the test duration, and the data quality rating. To simulate real data collection and smaller datasets, the order of the data points is randomised and the first part of the resulting dataset is evaluated up to 25 data points, up to 50 data points, and up to the full size of 515 data points. We do not examine dataset sizes between 50 and 515 data points because 50 data points on this dataset already performs almost as well as the full data, and we do not expect to see much improvement. We do not examine dataset sizes below 25 data points because this dataset size already yields scores close to the expected value of a random prediction, and the probability for the null hypothesis being true is already above 10% (see Table 3). A smaller sizes would not yield better results and this dataset size is a suitable example for a dataset whose trained classifier may not generalise to more data points. For dataset sizes between 25 and 50 data points, we believe that no additional insight would be gained, as the larger datasets contain the entirety of the smaller datasets. Furthermore, our results indicate that the 50 data point dataset is a good example for a dataset on which a classifier can be reliably trained on and getting close to its optimal performance despite not using many data points (see Tables 2 and 3). This is demonstrated by the comparison to the complete dataset. Random sampling from a more extensive population introduces varying degrees of skewness within different dataset sizes. The class ratio within the 25-data point dataset stands at 11:14; for the 50-data point dataset, it becomes 20:30; while for the comprehensive dataset, it shifts to 201:314. The BCWD dataset is a widely used benchmark dataset in ML, particularly in the field of classification. It consists of 569

samples, each representing a different patient, and 30 features describing characteristics of cell nuclei present in digitised images of fine needle aspirate (FNA) of breast mass. The feature measurements include mean values, standard error, and 'worst' or largest (mean of the three largest values) measures of radius, texture, perimeter, area, smoothness, compactness, concavity, concave points, symmetry, and fractal dimension of the cell nuclei. The outcome variable is a binary label indicating whether the FNA sample is malignant or benign. The class distribution across the sampled smaller datasets are as follows: 11:14 for the 25-sample subset, 15:35 for the 50-sample subset, and 212:357 for the complete dataset. The MNIST dataset is another popular benchmark dataset in ML, specifically for image recognition tasks. It contains 70,000 grayscale images of digits 0–9, each 28x28 pixels in size resulting in 784 features. The dataset is divided into 60,000 training images and 10,000 testing images. Each pixel value ranges from 0–255, where 0 represents black and 255 represents white. In alignment with the focus of this paper on investigating small datasets, we randomly extracted a subset of 250 data points from the complete dataset. From this reduced sample, we further derived two even smaller subsamples consisting of 50 and 25 data points respectively. Regrettably, evaluating the entire MNIST dataset fell outside the scope of our investigation. For this binary classification problem, the objective is to determine whether the image depicts a '1' or not. The subset of 250 data points contains 36 instances marked as '1'. Similarly, the 50-point subset includes 6 such instances, while the smallest subset of 25 data points encompasses merely 2 instances denoted as '1'. For the evaluation of all data one node consisting of 24 cores of the high performance computer 'Elwetritsch' at the University of Kaiserslautern-Landau (RPTU) was utilised. The code is completely written in Python and mainly makes use of the scikit-learn library [30]. It can be found at https://osf.io/8rkjb/.

## Preprocessing

We did not eliminate outliers for the CESAR data because it uses the self-assessment of data quality, which can be learned by the algorithm if it really has an impact on the predictions. In addition, the evaluation of this dataset is only exemplary. Therefore, we did not focus on optimising the prediction. Since some of the data used have widely varying ranges of values, the data must be normalised or standardised. In this case, the data is standardised because we do not eliminate the outliers and the influence of the outliers is much smaller with standardisation than with normalisation [8]. In addition, the SVM algorithm used does not rely on a fixed range of values between 0 and 1. The responses to the questions are categorised solely by numerical codes, with no additional significance attached to larger numbers. As such, all columns apart from those pertaining to the self-assessment of data quality and processing duration were subjected to one-hot encoding resulting in 73 features (Supporting information). For the MNIST and BCWD datasets, no one-hot encoding is necessary because they do not contain categorical data except the target variable. Of course, the self-generated data do not need to be cleaned of outliers, and the normalisation of these data would not be necessary, but the influence of the metrics and the method should be studied as close to practice as possible. Therefore, the data is standardised so that a possible influence of this procedure is included in the results and so that realistic results can be obtained.

## Feature selection

No feature selection was performed for the generated datasets, as they are generated so that all features have the same importance. For the CESAR, BCWD and MNIST datasets, the features were ranked using mutual information (see Supporting information) and the most important features were used, treating the number of features to be selected as a

hyperparameter to be adjusted. The mutual information implementation for categorical data from scikit-learn [30] was used. This method was selected because it is rather simple, fast, easy to implement, and—most importantly—non-parametric, which is sufficient to demonstrate the influence of feature selection. If the prediction quality is to be optimised and the choice of features is important, other more sophisticated methods are recommended (see the Supporting information).

## Handling class imbalances

For synthetically generated datasets specifically designed for balance, specialized techniques to manage class imbalance are unnecessary. This is because such considerations are inherently incorporated during the dataset creation process. However, for the real-world datasets that were examined, these techniques remain crucial due to the presence of class imbalance. For the CESAR, BCWD and MNIST datasets, we decided to use random oversampling because it is a simple, easy-to-understand method that is sufficient for the demonstration. However, random oversampling has a few disadvantages compared to more sophisticated methods (see the Supporting information), so we generally recommend using alternatives as mentioned in section Supporting information (e.g. SMOTE).

## Hyperparameter tuning

For both the synthetic and real world datasets, the best hyperparameters for each dataset are searched for during training. This is done using the GridSearchCV function of scikit-learn [30], which attempts all possible combinations of the given parameters and selects the one with the highest prediction quality. To determine the prediction quality, the function uses CV; in this case, we chose to use a stratified 5-CV, as it still yields good results but takes less time to compute than a 10-CV. Since the aim of this work is not to find optimal classifier performance but to demonstrate differences between using hyperparameter and feature selection, we tried to generate clear differences between the hyperparameters and number of features used. Therefore, we used for the C parameter of the SVM the default value of '1' as well as its value divided by ten and multiplied by ten. For 'gamma', we chose both available options for choosing a gamma value automatically as well as choosing at random the value of 0.1. We tested all in scikit-learn available kernels and tried a number of features ranging from 10 to its threefold of 30. The resulting parameter space under investigation is spanned by the following parameters 'C': 0.1, 1, 10; 'gamma': 0.1, 'scale', 'auto'; 'kernel': 'linear','rbf','poly','sigmoid'; 'n_features_to_select': 10, 15, 20, 25, 30. Here, 'C' is the regularisation parameter, which controls the importance of misclassifications. 'gamma' controls the width of the bell-shaped curve of the RBF kernel, and the value 'scale' sets gamma to the inverse of the product of the number of features and their variance. The value 'auto' sets the gamma to the inverse of the number of features. The parameter 'n_features_to_select' controls how many features must be selected during feature selection. 'kernel" controls the SVM kernel used, where 'linear' assumes linear separability of classes, 'rbf' uses the RBF kernel, and 'sigmoid' uses the sigmoid kernel. (see Supporting information). As mentioned, we employ a simplified approach that we deem adequate for our purposes, as our objective does not involve maximising predictive accuracy but demonstrating the method for real data. Nevertheless, should optimal results become the aim, implementing an automated hyperparameter optimisation could prove beneficial (see Supporting information).

## Metrics

Both the training and the evaluation of the prediction quality of a dataset are always performed with the same metric. We study the following metrics: $F_1$, recall, precision, ACC, balanced accuracy (BA), Cohen's kappa ($\kappa$), area under the receiver operating characteristic curve (AUC), and the Matthews correlation coefficient (MCC). We selected these metrics because they are commonly used in the literature and there are several papers that address their advantages and disadvantages (see Supporting information). Additionally to investigating these metrics on a large esemble of synthetical generated balanced datasets we exemplary investigate the differences between using MCC, ACC and $F_1$ for evaluating MNIST, BCWD and CESAR dataset. For the CESAR dataset are also recall, precision, BA, $\kappa$ and AUC demonstrated.

## Determination of the probability of the null hypothesis

In this paper, we use an empirical, non-parametric permutation test (see the Supporting information) to test whether two groups (classifier performance on original data and on data with randomly permuted labels) are equal (null hypothesis), because for this test the dataset does not need to fulfil any assumptions regarding the statistical properties. In this manner, the presented approach is more transferable to various small datasets that may not meet the statistical requirements of a parametric test. We believe that the computational time required for the test is still acceptable for small datasets and the methods studied here. The test randomly permutes labels and reassigns them to other attributes, thereby breaking the link between labels and attributes. In this work, only 25 permutations are performed for the synthetic datasets, and the performance of the ML algorithm is evaluated using both the rnCV and nCV methods. For the evaluation of the CESAR, MNIST and BCWD datasets, we use 50 permutations to resolve the difference in the returned probability between the datasets. In fact, 999 or more permutations are required for an effective approximation of the probability (see Supporting information). However, the focus of this work is on the qualitative behaviour and the demonstration of the evaluation methodology and, therefore, fewer permutations are used for reasons of computational time. For each permutation and each dataset, an rnCV is performed with five repetitions as well as five nested CVs with different random seeds. This is done to estimate the influence of randomness, to obtain a mean that is less dependent on randomness, and to examine the difference between using rnCV and simple nCV in permutation testing. The same permutations are used to calculate the probability of the rnCV as those used to calculate the nCVs, so we use their results to calculate the results of the rnCV. In addition, this allows for good comparability of results, as differences are entirely due to the calculation method and not due to chance.

## Validation methods

The predictive quality of a model is evaluated for each dataset using both an rnCV and an nCV.

**Nested cross-validation.** Depending on what you want to do with your data (e.g. use feature selection or not), the structure of the nCV used varies. The structure in our case is depicted in Fig 2. To avoid overly optimistic performance estimates and to increase comparability, particularly for asymmetric metrics, the minority class of the full dataset is always selected as the positive class. The outermost *stratified n-CV* divides the full dataset into n equal parts with class proportions that reflect the class proportions of the full dataset. Each of these parts is used once as test data to determine the prediction quality of the model trained in the following steps. To estimate the prediction quality, we choose for the synthetic and the CESAR

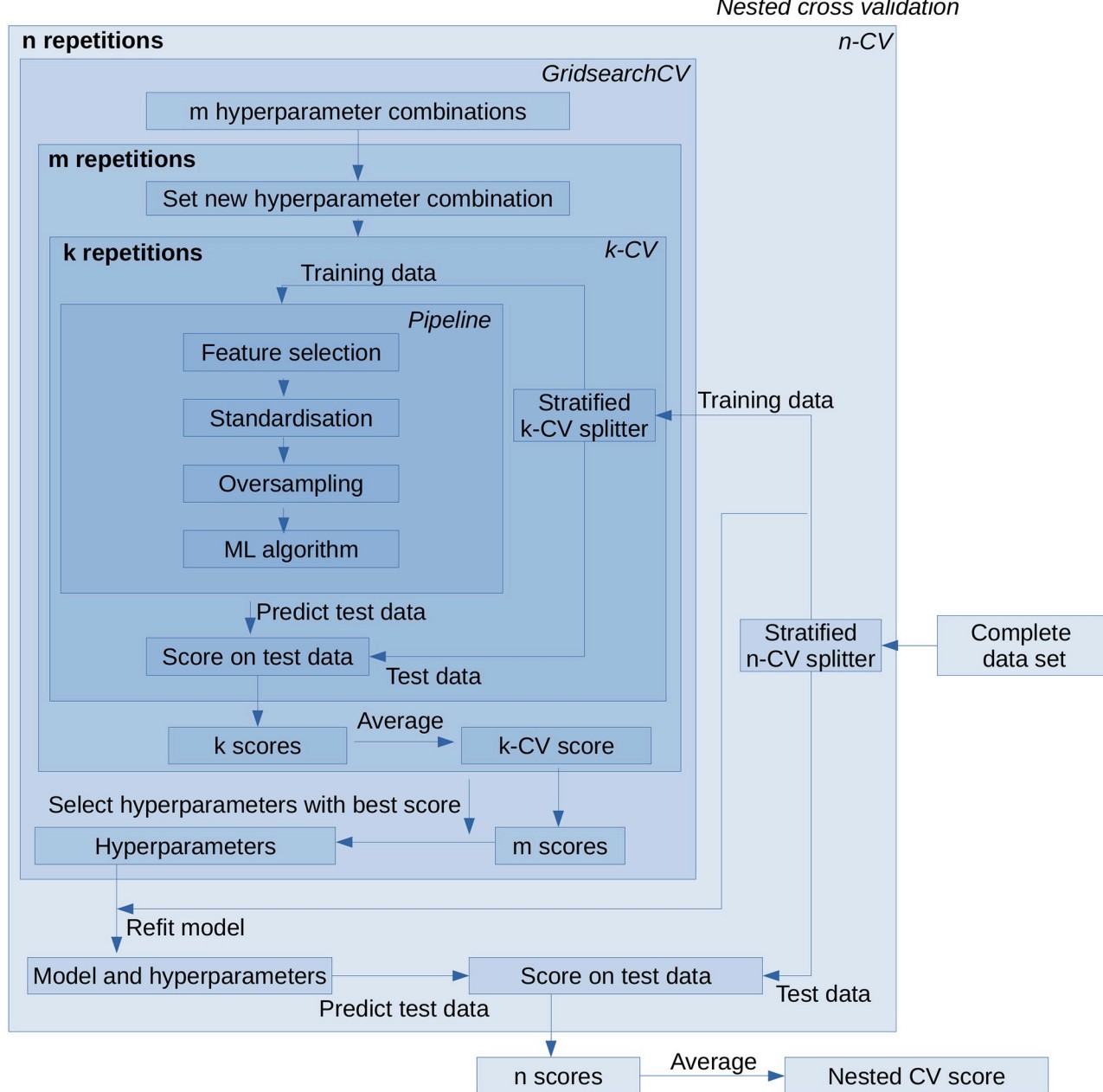

**Fig 2. Structure of the nested cross-validation method in this study.** It consists of an outer CV to evaluate the performance of the trained ML model and an inner CV to find the optimal (hyper)parameters and features of the model. In the innermost loop, feature selection, standardisation, and oversampling are applied to the training subset of the inner CV before passing the data to the SVM classifier. The combination of hyperparameters that achieves the highest score in the inner CV is used to retrain the model on the outer CV training data and predict its test data. The averaged results of all iterations of the outer CV are returned as the result.

datasets n = 10 for the outer stratified n-fold CV, as it yields results with low bias and variance (see the Supporting information). For the BCWD and MNIST datasets, we use the same parameters except that we used a 5-CV for inner and outer CV. The reason for this is the strong imbalance of the MNIST dataset regarding ones and all other numbers. For the dataset versions consisting of only 50 and 25 points there are not enough ones for every fold otherwise.

The other n-1 parts are passed to the *GridsearchCV* function of scikit-learn [30], which systematically uses all *m possible combinations of the given hyperparameters* (see section Hyperparameter tuning) and divides the training dataset passed from the outer n-CV into k parts with equal class ratios in another inner *stratified k-CV*. Each of these k parts is used once to test the prediction quality of the model created in the following steps, and the other k-1 parts are used to train the model. As a compromise between computational time and variance and bias of the results, we choose k = 5 in the inner stratified k-fold CV for model selection. Thereafter, the training dataset of the inner CV is passed to a *pipeline*, which *selects features* based only on this training dataset (see section Feature selection), then *standardises* the data (see section Preprocessing), balances the class ratios by *random oversampling* (see section Handling class imbalances), and finally passes the processed training data to the actual *ML algorithm*. For the ML algorithm, we chose an SVM because it is widely used and can handle small datasets, but other classifiers such as a random forest would also be suitable. The SVM trained on these k-1 parts is then applied to the last part of the inner CV, which is kept for testing; using a metric such as the MCC, the *prediction quality score* is determined by comparing the predictions with the true values. After all k parts have been used once as a test dataset, the *k scores* of the prediction qualities are averaged and thus the prediction *score of the k-CV* is determined. This is done for all m combinations of hyperparameters. From the resulting *m scores*, the combination with the highest score is selected. The selected model is *refitted* to the full training dataset of the n-CV and used to predict the corresponding test dataset. After each of the n parts is used as a test dataset, the obtained *n scores* are averaged and thus the final *score of the nCV* is obtained.

**Repeated nested cross-validation.** The nCV method described above is repeated five times with different random seeds and the results are averaged to obtain the result of the rnCV.

**Train/test split.** For a Train/Test Split, a portion of the data is used to train the classifier and another part is utilised to test the fully trained classifier. To demonstrate the variance of this approach on small datasets, we evaluate the full BCWD dataset five times using this method with different random seeds using ACC. In each instance, 80% of the data is designated as the training dataset, while 20% of the data served as the testing dataset.

## Ablation study

The main goal is to identify which elements have the most significant impact on overall performance of the rnCV and understand their respective roles better. To achieve this, we identified the most important components of our rnCV and evaluated our real world data multiple times. In each iteration, one component was removed while keeping all other conditions constant. This allowed us to isolate the effects of removing specific components without introducing additional confounding factors. We evaluated the full rnCV, the rnCV without feature selection, the rnCV without hyperparameter tuning as well as the rnCV with neither hyperparameter tuning nor feature selection (therefore not using the inner CV). Because the MCC is most suitable for skewed data (Supporting information) we use the MCC when investigating the ablation of rnCV components.

## Confusion matrix

The Confusion Matrix is a table that is often used to describe the performance of a classifier on a set of test data for which the true values are known. It provides a way to visualize the number of false positives, false negatives, true positives, and true negatives [8]. In the context of our study, the first row of the Confusion Matrix corresponds to the actual members of the first category, while the second row corresponds to the actual members of the second category. The

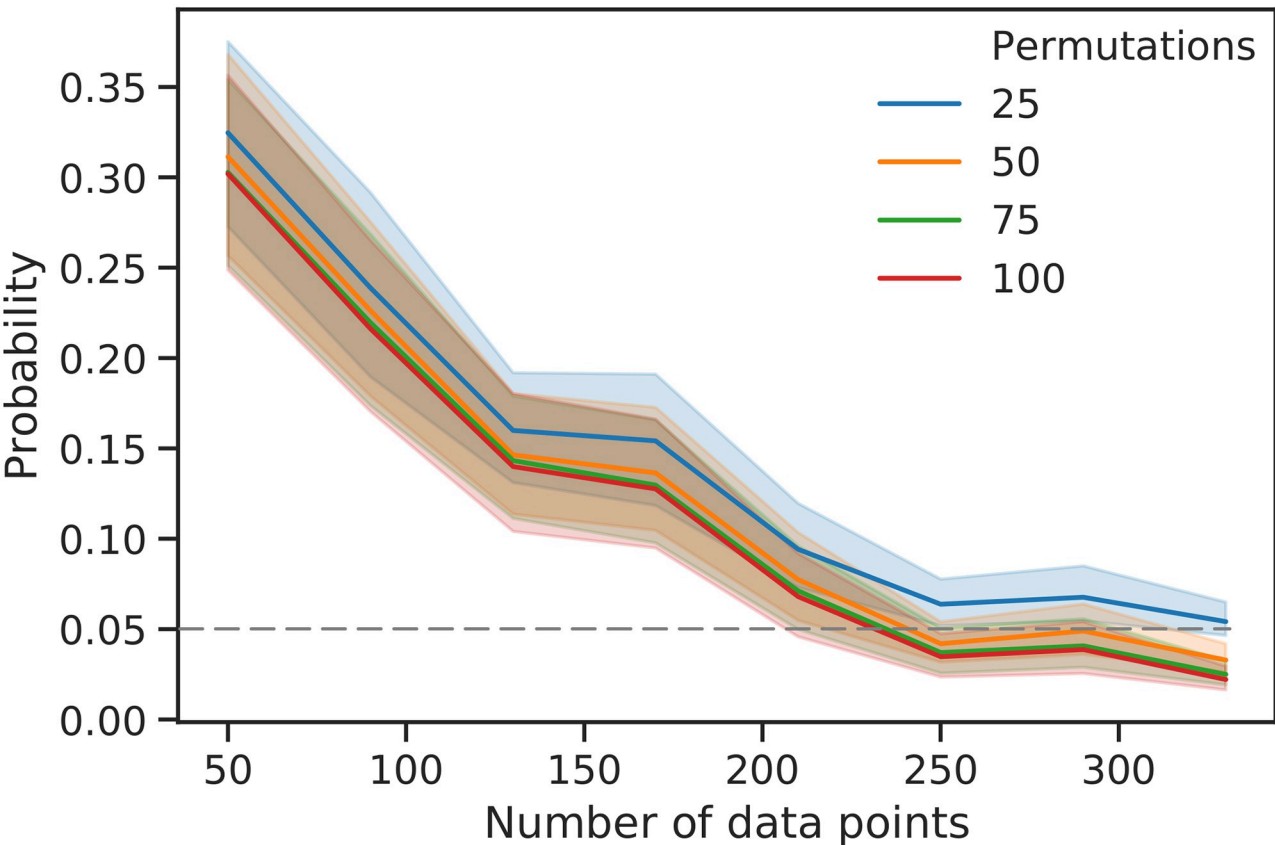

**Fig 3. Dependence of the probability obtained with the permutation test using the MCC on the number of data points for different numbers of permutations.** The probability obtained is depicted as a function of the number of data points for 25 (orange), 50 (blue), 75 (red), and 100 permutations (green). The coloured area represents the 95% confidence interval calculated over all datasets using bootstrapping—that is, it is derived from the results of repeated resampling of the population. The dotted grey line indicates a probability of 5%.

first column represents the members which are predicted as belonging to the first category, while the second column represents those predicted as belonging to the second category. Since the rnCV method produces multiple estimators and consequently multiple confusion matrices, we will examine the average of all generated matrices. This approach allows us to better understand the overall performance of our classifiers across different iterations.

### Artificial intelligence tools and technologies

For some paragraphs of this paper, the authors utilised 'DeepL Write' as well as 'Mixtral 8x7B' for the sole purpose of improving the language of already written text while carefully checking that the content wasn't altered by these AI tools.

## Results

### Influence of the number of permutations on the result of the permutation test

We determined the probability that the null hypothesis is true for balanced synthetic datasets using a permutation test using rnCV and the MCC (Fig 3). At a sample size of 50 data points, the probability is between 0.30 and 0.33 for each number of permutations ($N$ = 25, 50, 75,

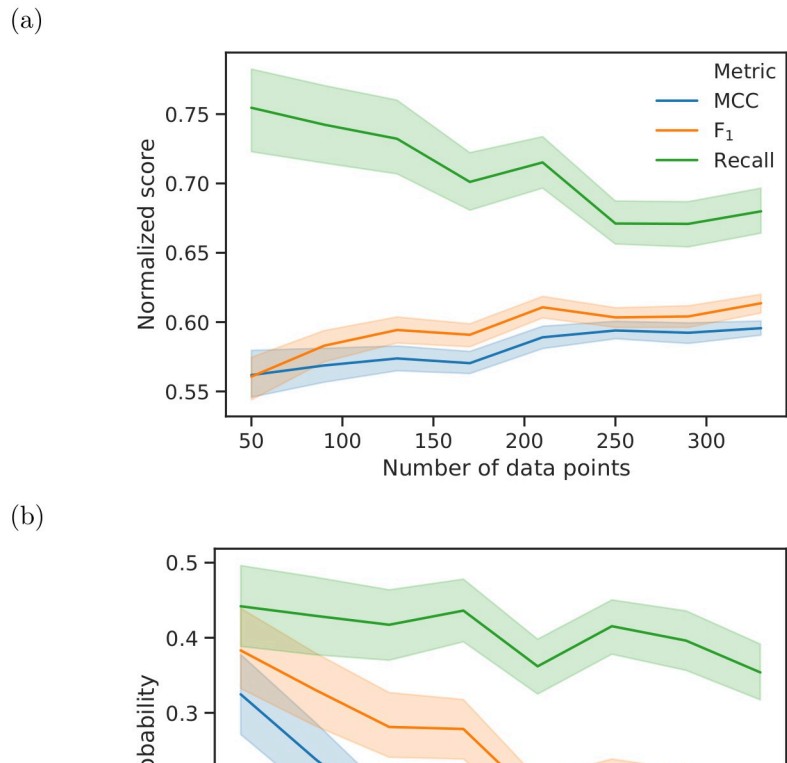

**Fig 4. (a) Dependence of the normalised prediction score and (b) the probabilities of the corresponding permutation tests on the number of data points using MCC, $F_1$ and recall.** The MCC score was normalised to the same range of values from 0 to 1 as the other two metrics. The coloured area represents the 95% confidence interval calculated by bootstrapping over all datasets.

100). The average probabilities decay with an increasing number of data points, and they cross a probability of 0.05 at approximately 230 data points ($N = 100$), at 235 data points ($N = 75$), and 240 data points ($N = 50$). The examined range up to 330 data points is not sufficient for 25 permutations to attain an average probability of less than 5%.

## Comparison of metrics

A comparison of three common evaluation metrics reveals that the the average MCC increases slightly with increasing number of data points (Fig 4), while the probability obtained by the permutation test decreases (Fig 4). The MCC has a lower probability of the null hypothesis being true than the $F_1$ score and the recall for all data points. Although the increase in the $F_1$ score with more data points is greater than the increase in the MCC, there is a slower decrease in its probability. If we examine the dependence of the normalised recall score on size of the dataset (Fig 4), the recall decreases as the number of data points increases. However, the recall

shows values above 0.65, which is at the upper end of its value range. Despite the decrease in recall, the probability obtained by the permutation test (Fig 4) nevertheless declines when there is a larger number of data points. The probability of recall is higher than that for the other metrics for all numbers of data points. Comparing MCC, $\kappa$, ACC, BA (S1 Fig), the trends are very similar and there are only very small deviations among them. As the amount of data increases, the 95% confidence interval decreases for all metrics, but this decrease is most evident for the MCC.

## Comparison of nCV and rnCV

To understand the influence of repetitions of the nCV, we averaged the results of all equally sized balanced synthetic datasets for each number of data points. We used five repetitions within the rnCV to determine the MCC and the probability, and obtained the results for MCC and probability by averaging the results of five nCVs. This means that we used the same predictions to calculate the results for the nCVs and that of the rnCV; thus, only the method of calculation differs and, therefore, the results for this comparison are exact.

It must be noted that the score of the rnCV (see Fig 5) is identical to the mean score of the nCVs. However, the probabilities of the null hypothesis being true are consistently higher for the nCV than for the rnCV (see Fig 5).

To demonstrate the improved reliability of the rnCV compared to the nCV, we used a similar method to evaluate the CESAR dataset as well as its subsets. Table 1 shows the minimum and maximum MCC score of each of the five nCVs and their standard deviation. It also shows the probability of using a single rnCV compared to the averaged probability of the nCVs. As with the balanced synthetic data, the probabilities are higher when only nCVs are used. In addition, the standard deviation and the difference between the minimum and maximum score achieved increase with fewer data points.

## Real world datasets: Comparison of metric score and null hypothesis probability between metrics

To quantify the impact of this procedure on a real dataset, we determined different metrics and the probability that the null hypothesis is true for the publicly available datasets, CESAR, BCWD and MNIST. The probabilities were calculated for 25 and 50 random data points of the dataset as well as for the complete dataset. In this manner, we effectively simulate results of smaller real datasets with comparable data quality. Tables 2 and 3 present the results of an complete rnCV.

For all investigated complete real world datasets all metrics are well above the expected value of a random data prediction (see Table 2 and S3 Table) and the permutation tests indicate that the results are statistically significant—that is, $p < 0.05$ (see Table 3 and S4 Table). The scores of the metrics of the MNIST and BCWD don't differ much and are both notably higher than the scores of the CESAR dataset. Despite the fact that the full dataset is much larger than its 50 data point variant, the metric values for the 50 data point CESAR dataset are for BA, precision, $F_1$ and $\kappa$ only slightly lower than for the full dataset and for the other metrics even slightly higher or the same. For the MNIST dataset the MCC and $F_1$ score do noticeably degrade compared to the complete dataset while the ACC only decreases slightly. Nevertheless they are still far above chance value and the probability for the null hypothesis is only 2%. For 25 data points of the CESAR and MNIST datasets the metrics are barely above and in some cases below the expected value of chance; the probabilities are not statistically significant. For the BCWD dataset, the probability of the null hypothesis is at 2% for all dataset sizes for ACC, $F_1$ and MCC. It is therefore the only dataset in our test, where 25 data points are enough to

(a)

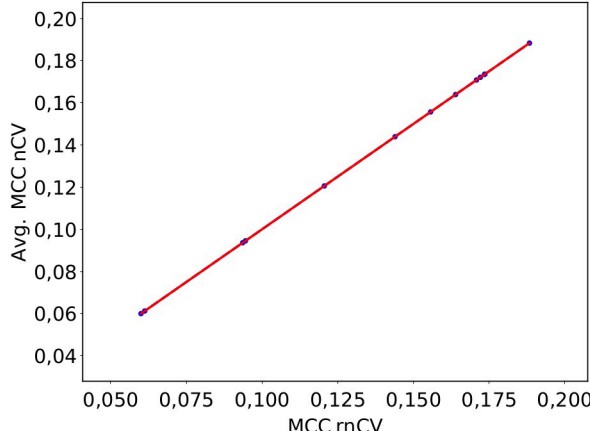

(b)

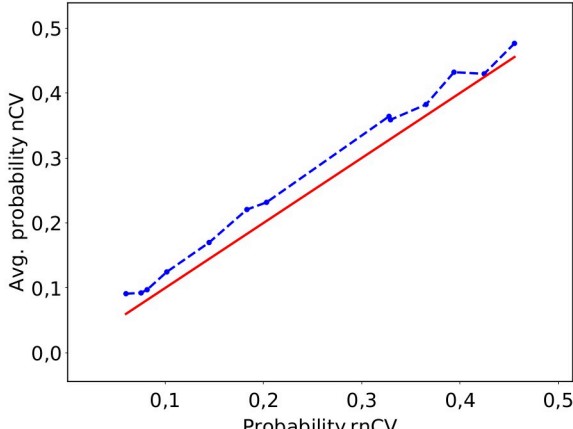

**Fig 5. Mean value of (a) the MCC, and (b) the probability of 5 nCV versus the MCC of an rnCV consisting of five replicates.** The red line represents the bisector for clarity and the connecting line between the data points is only a guide for the eye.

train a classifier for which we can reject the null hypothesis. As examples, we analysed the errors made by the classifiers trained on ACC, F1 and MCC. Table 4 shows that the classifier trained on the ACC for the complete CESAR dataset as well as the 50 data point dataset slightly prefers to predict the majority class, and therefore, more often misclassifies a minority class

**Table 1. Minimum, maximum and standard deviation of scores of the MCC for the individual nCV components of a rnCV.** It includes the probability of the null hypothesis when evaluating using rnCV, contrasted with the average probability derived from utilising nCV for five iterations in the permutation test, rather than a singular rnCV. Assessments are performed on random subsets of the CESAR dataset.

| Points | MCC Min | MCC Max | MCC standard deviation | Probability rnCV | Averaged probability nCVs |
|---|---|---|---|---|---|
| 515 | 0.32 | 0.40 | 0.03 | 0.02 | 0.02 |
| 50 | 0.29 | 0.52 | 0.09 | 0.04 | 0.05 |
| 25 | 0.00 | 0.40 | 0.16 | 0.15 | 0.35 |

**Table 2. Scores of the ACC, BA, precision, recall, $F_1$-Score, MCC, AUC, and $\kappa$ for rnCV on a random subsets of the CESAR dataset.**

| Points | ACC | BA | Precision | Recall | $F_1$ | MCC | AUC | $\kappa$ |
|---|---|---|---|---|---|---|---|---|
| 515 | 0.70 | 0.67 | 0.70 | 0.90 | 0.63 | 0.36 | 0.74 | 0.36 |
| 50 | 0.72 | 0.68 | 0.67 | 0.99 | 0.59 | 0.40 | 0.74 | 0.35 |
| 25 | 0.56 | 0.45 | 0.45 | 0.94 | 0.41 | 0.16 | 0.54 | 0.13 |

**Table 3. Probabilities of the ACC, BA, precision, recall, $F_1$-Score, MCC, AUC and $\kappa$ for rnCV on a random subsets of the CESAR dataset.**

| Points | ACC | BA | Precision | Recall | $F_1$ | MCC | AUC | $\kappa$ |
|---|---|---|---|---|---|---|---|---|
| 515 | 0.02 | 0.02 | 0.02 | 0.02 | 0.02 | 0.02 | 0.02 | 0.02 |
| 50 | 0.02 | 0.02 | 0.02 | 0.02 | 0.04 | 0.04 | 0.02 | 0.02 |
| 25 | 0.29 | 0.31 | 0.12 | 0.12 | 0.45 | 0.15 | 0.39 | 0.19 |

**Table 4. Confusion matrices of the ACC, $F_1$-Score, and MCC for rnCV on a random subsets of the CESAR dataset.**

| Points | ACC | $F_1$ | MCC |
|---|---|---|---|
| 515 | $\begin{pmatrix} 24.8 & 6.6 \\ 8.8 & 11.3 \end{pmatrix}$ | $\begin{pmatrix} 19.5 & 11.9 \\ 5.5 & 14.6 \end{pmatrix}$ | $\begin{pmatrix} 22.2 & 9.2 \\ 7.0 & 13.1 \end{pmatrix}$ |
| 50 | $\begin{pmatrix} 2.5 & 0.5 \\ 0.9 & 1.1 \end{pmatrix}$ | $\begin{pmatrix} 2.1 & 0.9 \\ 0.7 & 1.3 \end{pmatrix}$ | $\begin{pmatrix} 2.4 & 0.6 \\ 0.9 & 1.1 \end{pmatrix}$ |
| 25 | $\begin{pmatrix} 0.8 & 0.6 \\ 0.5 & 0.6 \end{pmatrix}$ | $\begin{pmatrix} 0.6 & 0.8 \\ 0.5 & 0.6 \end{pmatrix}$ | $\begin{pmatrix} 0.8 & 0.6 \\ 0.5 & 0.6 \end{pmatrix}$ |

class member as majority class member than the other way around. For the 25 data point dataset, the prediction is quite balanced. The F1 score on the other hand seemed to more often misclassify a majority class member as minority class member for all investigated dataset sizes. For the MCC, there is not such a clear trend visible. For the complete dataset and 25 data points, it slightly more often misclassifies a majority class sample as minority class, for 50 data points, it is the other way around. For the MNIST and BCWD dataset, there are no big difference in confusion matrices between different metrics (see S5 Table).

## Real world datasets: Ablation study

In the case of the CESAR dataset, the full dataset exhibited comparable performance with or without hyperparameter tuning and feature selection (see Table 5). With 50 data points, leaving out either feature selection or hyperparameter tuning resulted in degraded metric scores compared to including both components. Using neither feature selection nor hyperparameter tuning resulted in the worst performance. Interestingly, for 25 data points leaving out feature

**Table 5. Scores of the MCC for rnCV missing either the hyperparameter tuning, the feature selection or both on a random subsets of the CESAR dataset.**

| Points | full rnCV | missing feature selection | missing hyper parameter tuning | missing feature selection and hyper parameter tuning |
|---|---|---|---|---|
| 515 | 0.36 | 0.35 | 0.38 | 0.37 |
| 50 | 0.40 | 0.30 | 0.37 | 0.18 |
| 25 | 0.16 | 0.19 | 0.25 | -0.02 |

**Table 6. Probabilities of the MCC for rnCV missing either the hyperparameter tuning, the feature selection or both on a random subsets of the CESAR dataset.**

| Points | full | missing feature selection | missing hyper parameter tuning | missing feature selection and hyper parameter tuning |
|---|---|---|---|---|
| 515 | 0.02 | 0.02 | 0.02 | 0.02 |
| 50 | 0.04 | 0.04 | 0.04 | 0.15 |
| 25 | 0.15 | 0.20 | 0.15 | 0.57 |

selection or hyperparameter tuning increased the score, but leaving out both decreased it. The corresponding probabilities in this case on the other hand got worse or stayed the same despite increasing scores (see Table 6). In case of the MNIST dataset, the ablation of the feature selection improved the scores as well for all dataset sizes if left out alone or if left out together with hyperparameter tuning (see S6 Table). Leaving out hyperparameter tuning did not change the score notably. In general, the classifier seems to perform very well on both the 250 data point subset as well as on the 50 data point subset. Only in case of 25 data points the MCC drops to zero when using feature selection, and the performance without feature selection is strongly diminished compared to using more data. The probability is always 0.02 except when using feature selection, where it becomes 1.0 for 25 data points (see S7 Table). All probabilities of the BCWD dataset achieve the minimal value of 0.02 for all dataset sizes and ablations, because high MCC scores are yielded in every case. Nevertheless there are differences between the scores. While it does not seem to make a big difference when ablating any component or both for the full dataset, it gets more impact for smaller dataset sizes. In case of 25 data points, leaving out feature selection degrades the score by 0.27 while leaving out only hyperparameter tuning leads only to a marginal degradation. Curiously leaving out both degrades the score much less than only leaving out the feature selection.

## Discussion

For the balanced synthetic datasets a smaller number of permutations implies that the permutation test overestimates the probability that the null hypothesis is true. Thus, fewer permutations are more likely to lead to false acceptance of the null hypothesis than to false rejection. However, the qualitative relationship between the probability and the number of data points appears to be preserved; thus, a small number of permutations is sufficient to make qualitative statements regarding their relationship. In general, at least 999 permutations should be selected for an effective approximation of the true probability according to Edgington [24] (see Supporting information). In line with classical statistics, we suggest a probability threshold of 5% that the null hypothesis is true—that is, to consider the result of a classification task to be significant. In this study, this value proved to be useful to interpret the results of the smaller and larger datasets. We used the same original and permuted datasets to analyse the characteristics of different metrics. The comparison of the dependence of the recall and the corresponding probability (Fig 4) on the dataset size confirms that a high score alone does not imply a relationship between features and target variables. This is also evident in the only moderate increase of the MCC with a simultaneous strong decrease in the associated probability as well as in the narrowing of the confidence interval of all metrics when using more data points. A narrowing confidence interval implies that it becomes increasingly unlikely to obtain values much larger or much smaller than the mean. This is consistent with the decreasing probability of the permutation test, whereas the score of the metric remains the same or even decreases. Thus, the score alone is not sufficient and a permutation test should be

performed in any case. In addition, the result of the permutation test also depends on the size of the dataset.

The synthetic datasets studied are somewhat predictable, since they are generated with a small difference in the distribution of the features of both classes. In other words, the null hypothesis can be rejected for these data, and the metrics should provide the lowest possible probability for it. From the much lower probabilities obtained by performing a permutation test using the MCC compared to those obtained using the recall, it is evident that the MCC is better at detecting a relationship between the features and the target variable than the recall. This is also true for the $F_1$ score calculated from recall and precision. $\kappa$, MCC, ACC, and BA are equally suitable for the evaluation of the studied datasets. However, one should keep in mind that previous research [31] considers MCC more suitable and ACC rather unsuitable for unbalanced datasets, which are only exemplary examined here (see S1 and S2 Tables). It should be noted that the high probabilities >5% of the null hypothesis being true for all metrics are partially due to the small number of features used and the small difference in the value distributions of the two classes. With more features and a larger difference between the classes, better probabilities could be obtained with fewer data points.

The equivalence of the average nCV score in Fig 5 to the rnCV score can be attributed to their computation from identical predictions using the same process. Nevertheless, the nCV probabilities generally surpass those of rnCVs due to differing computational methods: The nCV probability is derived by comparing a single nCV from the original dataset to corresponding single nCVs within the permuted dataset, yielding an empirical probability. This approach can lead to slight underestimation of the original dataset score and potential overestimation of the permutation score due to nCV variability. Conversely, the rnCV calculates the mean nCV scores before comparing them against equivalent permutation means, reducing result variability. As a result, individual permutations are less likely to outperform or underperform the original dataset, enhancing overall result reliability for rnCVs relative to nCVs. This improved reliability decreases the likelihood of incorrectly concluding that the null hypothesis is true when it is actually false. It is crucial to highlight that this direct comparison is feasible owing to consistent permutation usage across both nCV and rnCV calculations, as well as the rnCV probability computation relying on nCV outcomes.

When comparing the metrics used in the analysis of three distinct datasets—MNIST, BCWD, and CESAR—it was observed that the scores were significantly higher for MNIST and BCWD than for CESAR at all dataset sizes (see Table 2 and S3 Table). Furthermore, the confusion matrices showed minimal differences across various metrics for MNIST and BCWD (see S5 Table) and more significant ones for the CESAR dataset (see Table 4). All this suggests a stronger link between features and labels in MNIST and BCWD than in the CESAR dataset. There, the values of the metrics decreased significantly with smaller dataset sizes, and more significant differences in the confusion matrices were observed between the use of different metrics (see Table 4).

The value of repeatedly assessing small data sets like CESAR that challenge a classifier's prediction capabilities is showcased in Table 1. Had only one repetition been used for a set of 50 data points, the MCC obtained could have been 0.52, and it would be possible that the MCC obtained from the full set would have been 0.32. This might foster a misunderstanding about the classifier's efficacy with larger data sets. Moreover, employing rnCV during the evaluation of probability, rather than simply averaging probabilities from several nCVs, holds significance. If we used multiple nCVs, it might indicate non-significant ($p > 5\%$) classification results for 50 data points; yet, applying rnCV establishes statistical significance. The variance and therefore the importance of using repetitions increases with smaller dataset sizes. Furthermore, the 50 data point subset, although significantly smaller than the full CESAR dataset,

shows only slightly inferior or even slightly superior scores, depending on the metric used (see Table 2), which might initially imply comparable classifier performance. However, a closer examination of the probability distributions reveals slightly poorer outcomes for MCC and $F_1$, indicating that some of these scores could be influenced by the specific data structure rather than the model's effectiveness (see Table 3). Consequently, this highlights the importance of considering factors beyond mere absolute scores for accurately evaluating a dataset's predictability.

For the CESAR and BCWD datasets (see Table 5 and S6 Table), the MCC scores got lower or at least remained the same when leaving out hyperparameter tuning or feature selection which is expected as parameters are chosen which are better suited to predict the data and not relevant features are ignored. This is not the case for the 25 data point CESAR dataset as the scores increase when leaving out either of both but heavily degrade when leaving out both at once. This seems less strange when looking at the corresponding probabilities (see Table 6). The probabilities get worse or in best case stay the same when leaving out hyperparameter tuning and/or feature selection. This indicates that by pure chance the datastructure of the 25 data point set was more suitable to be evaluated with the default parameters and features on all folds even when the classifier did only find a connection between features and classes which does not generalise. In this case, the use of the permutation test was necessary to fully understand the results. For the MNIST dataset, the scores always increase whenever no feature selection is used (see S6 Table). We suspect that the reason for this is that our search space for the features was inappropriate. We selected a maximum of 30 features out of the 784 available, which does not seem to be sufficient as indicated by the decreased score when using feature selection. This demonstrates that it is crucial to search for ideal hyperparameter and features using a suitable method and search space.

For the BCWD and CESAR datasets, the scores of the 50 data points are more or less the same as for the full dataset, although they consist of a multiple of data points (see Table 2 and S3 Table). Usually one would expect a classifier to perform better when trained on more data [9] so this may indicate that the classifier is limited by the intrinsic connection between features and labels in the investigated datasets. The scores of the BCWD dataset are in general much higher than for the CESAR dataset indicating that this is no effect caused by a high score consolidating the assumption that the inherent data structure prohibits much better predictions. Consistent with this interpretation, the MCC score differences between 50 data points and the full dataset are also small for the MNIST dataset when the feature selection of only 30 features is omitted.

The probabilities for the 25 data point subset of the BCWD and CESAR datasets are heavily dependent on the metric used for training and evaluation. Which metric performs best seems to depend on the dataset. For the BCWD dataset, the probability of the $F_1$ score is lower than for the ACC but for the CESAR dataset it is the other way around. It follows that it may be worthwhile to examine more than one metric, as they may happen to perform particularly well or poorly on the dataset. For not well predictable data, it again corroborates the usefulness of the permutation test which enables this comparison in the first place. With the highest probability of 2%, given the number of permutations used, the ML estimator uses the correlation between features and labels within the full CESAR dataset to predict the answer to question 5b. This prediction exceeds the expected value of zero associated with random classification, and significantly outperforms the worst case of -1, as indicated by a Matthews Correlation Coefficient (MCC) of 0.36. The similar prediction quality on only 10% of the data suggests that the dependence between questions is not sufficiently large to achieve much better accuracy. We were furthermore able to demonstrate our method of evaluation for the MNIST and BCWD datasets. We achieved an MCC score of 0.88 in predicting whether a cell sample is benign for

the full BCWD dataset with the lowest possible probability of the null hypothesis being true in our case. The largest subset of the MNIST we evaluated consisted of 250 data points, and the estimator achieved an MCC score of 0.89 in predicting whether an image is a one or not. It is important to note that these values are the average of the rnCV, and some individual nCVs yielded higher scores by chance. This again emphasises the value of the repetitions. Krstajic et al. [12] demonstrated the relevance of repeating evaluation with various random seeds in their work, Vabalas et al. [5] highlighted the importance of using nCV, and Ojala & Garriga [13] showcased the advantage of permutation tests. We have now shown that for optimal results on small datasets, all three methods must be combined. The use of nCV alone leads to higher variance, and particularly for small datasets when the estimator yields scores close to the expected value of random prediction further evaluation using a permutation test is necessary. A practical example where the use of repetitions could be interesting is the work by Naji et al. [18]. They used single train/test splits to evaluate BCWD and compare five ML algorithms. In our tests across five different train/test splits using a SVM, we obtained the worst accuracy as 0.89 and the best as 0.96. This fluctuation range is greater than the differences between the compared classifiers (0.94 to 0.97) in their work. An example where the use of the permutation test could have been advantageous is the work of Kılıç et al. [17]. Using binary classification on a dataset of 700 data points, they reported an ACC of 0.62. Given the proximity to the expected value of 0.5 from random prediction in balanced datasets, assessing the probability of the null hypothesis would be beneficial due to the variance of these results and uncertainty about which value would emerge through permutation of target variables.

## Conclusion

In social sciences, ML has a great potential to stimulate significant advances. When the aim is to evaluate the strength of the dependency between features and target variable and not to find one specific classifier, it is important to specify a probability value for the null hypothesis, particularly for small datasets and moderately good metrics. In this case, there may be a non-negligible probability that the classifier will fail to predict new data, as its performance is only an artefact of the data structure of the dataset under consideration. To obtain this probability, we suggest using a non-parametric permutation test in combination with a rnCV approach, which yields more reliable results than the traditional nCV method. Our results on synthetic data, where a low probability is expected, reveal that the probability determined by the permutation test is higher when the recall or $F_1$ score is used, compared to the MCC for our dataset. Furthermore, we found that evaluating datasets with multiple metrics can help to better assess the predictive quality of the model, as revealed by the analysis of the CESAR, MNIST and BCWD datasets. We demonstrate that repeating a nCV with different data splits reduces the variance of estimated classifier performance as well as the probability of randomly good results. We recommend using multiple repetition in the future evaluation of small datasets to reduce the variance especially if comparing results with only small differences. Most reliable results can be achieved by using the repetitions directly in the permutation test instead of running multiple tests and averaging afterwards. Furthermore, we recommend using the MCC as well as other metrics for training and evaluation, as the optimal metric may depend on the dataset.

## Supporting information

**S1 Fig. (a) Dependence of the prediction score as well as (b) the probabilities of the associated permutation tests on the number of data points using MCC, $\kappa$, ACC, and BA.** The scores of MCC and Cohen's Kappa were normalised to the same range of values from 0 to 1 as

the other two metrics. The coloured area represents the 95% confidence interval calculated by bootstraping over all datasets.
(TIF)

**S1 Table. Overview of the characteristics of different metrics.** In this table, the advantages and disadvantages of ACC, BA, $\kappa$, and $F_1$ Score are presented.
(PDF)

**S2 Table. Overview of the characteristics of different metrics.** In this table, the advantages and disadvantages of precision/positive predictive value (PPV), recall, AUC and MCC are presented.
(PDF)

**S3 Table. Scores of the ACC, $F_1$-Score, and MCC for rnCV on a random subsets of the MNIST and BCWD dataset.**
(PDF)

**S4 Table. Probabilities of the ACC, $F_1$-Score, and MCC for rnCV on a random subsets of the MNIST and BCWD datasets.**
(PDF)

**S5 Table. Confusion matrices of the ACC, $F_1$-Score, and MCC for rnCV on a random subsets of the MNIST and BCWD datasets.**
(PDF)

**S6 Table. Scores of the MCC for rnCV missing either the hyperparameter tuning, the feature selection or both on a random subsets of the MNIST and BCWD dataset.**
(PDF)

**S7 Table. Probabilities of the MCC for rnCV missing either the hyperparameter tuning, the feature selection or both on a random subsets of the MNIST dataset.**
(PDF)

## Author Contributions

**Conceptualization:** Steffen Steinert, Stefan Küchemann.

**Formal analysis:** Steffen Steinert.

**Investigation:** Steffen Steinert, Verena Ruf, David Dzsotjan, Nicolas Großmann, Albrecht Schmidt, Jochen Kuhn, Stefan Küchemann.

**Methodology:** Steffen Steinert, Stefan Küchemann.

**Software:** Steffen Steinert.

**Supervision:** Albrecht Schmidt, Jochen Kuhn, Stefan Küchemann.

**Visualization:** Steffen Steinert.

**Writing – original draft:** Steffen Steinert, Stefan Küchemann.

**Writing – review & editing:** Verena Ruf, David Dzsotjan, Nicolas Großmann, Albrecht Schmidt, Jochen Kuhn, Stefan Küchemann.

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
