## [Decision Letter · Decision Letter 0]

21 Nov 2023

PONE-D-23-36197A refined approach for evaluating small datasets via binary classification using machine learningPLOS ONE

Dear Dr. Steinert,

Thank you for submitting your manuscript to PLOS ONE. After careful consideration, we feel that it has merit but does not fully meet PLOS ONE’s publication criteria as it currently stands. Therefore, we invite you to submit a revised version of the manuscript that addresses the points raised during the review process.

We look forward to receiving your revised manuscript.

Kind regards,

Sathishkumar Veerappampalayam Easwaramoorthy

Academic Editor

PLOS ONE

Journal Requirements:

3. Please upload a copy of Supporting Information Figure/Table/etc. Supporting information S1 table, S2 table, S3 table and S4 table which you refer to in your text on pages 11 and 12.

Reviewers' comments:

Reviewer's Responses to Questions

**Comments to the Author**

1. Is the manuscript technically sound, and do the data support the conclusions?

Reviewer #1: Yes

Reviewer #2: Partly

2. Has the statistical analysis been performed appropriately and rigorously? 

Reviewer #1: No

Reviewer #2: Yes

3. Have the authors made all data underlying the findings in their manuscript fully available?

Reviewer #1: Yes

Reviewer #2: Yes

4. Is the manuscript presented in an intelligible fashion and written in standard English?

Reviewer #1: Yes

Reviewer #2: Yes

5. Review Comments to the Author

Reviewer #1: The authors presented a method for evaluating binary classification problems with small datasets. The problem is highly significant , but I have few concerns that need to be addressed.

1. The authors have not clearly specified the background of the problem and main contributions of the work.

2. The introduction section need to be properly organized with more clarity.

3. The experiment needs more regorios analysis. Moreover, hypertuning experiment as well as Ablation study can be included.

4. I suggest to add an additional experiment via error analysis that is usually helpful for model trst8ng with diverse conditions.

5. Have authors considered model evaluation via cross-domains? My suggestion is do more experimentation through cross- domain with small datasets.

6. Finally, use a real-time case of both images and other datasets to measure efficacy of proposed method

Reviewer #2: Dear Authors,

The manuscript "A refined approach for evaluating small datasets via binary classification using machine learning" presents an interesting topic by suggesting the use of a pre-processing of small datasets before using the Support Vector Machine Learning technique.

In this context, another widespread non-parametric technique is Random Forest. It would be interesting to mention this technique in the introduction chapter and highlight the reason for choosing the SVM algorithm.

To help you improve your manuscript, I offer the following recommendations:

1) It is necessary to review punctuation, especially the use of commas. Revise long paragraphs as they tend to confuse the reader. I've pointed out in the comments on the digital file the passages where the writing should be revised.

2) When acronyms are presented for the first time in the text, their meaning should be presented.

3) Highlight the objectives of the study in chapter 1.

4) Lines 62 to 85: a flowchart would illustrate the methodology used more effectively.

5) Line 106: "We do not examine dataset sizes below 25 data points because the performance is already not good for 25 data points". Please detail this restriction.

6) Line 153: Were the SVM parameters determined empirically? Please detail.

7) Line 188: "However, the focus of this work is on the qualitative behavior and the demonstration of the evaluation methodology and, therefore, fewer permutations are used for reasons of computational time." The parameter computational time appears repetitively in the text, making it clear that fewer permutations were chosen at the expense of less computational time. In this case, the use of cloud computing could solve this restriction.

8) Detail the applications used, as well as the hardware used.

9) The chapter reserved for discussions explores the results achieved. However, when compared with the results obtained by the authors cited in the references, the superior results achieved by the proposed methodology could be better substantiated.

10) Line 432: it would be interesting to create a chapter on conclusions. What are your recommendations for future studies?

I end my review by congratulating you on your study.

Respectfully,

6. PLOS authors have the option to publish the peer review history of their article (what does this mean?). If published, this will include your full peer review and any attached files.

Reviewer #1: **Yes: **Arvind Selwal

Reviewer #2: **Yes: **Marcos Benedito Schimalski, Professor at Santa Catarina State University, Brazil.

---

## [Author Response · Author response to Decision Letter 0]

4 Mar 2024

Dear reviewers and editor,

we would like to thank you for your helpful feedback to improve the quality of our paper. We have substantially revised the manuscript to follow your suggestions and hope that the current state is to your satisfaction.

Below we would like to address the points you raised.

Academic editor:

1. Please ensure that your manuscript meets PLOS ONE’s style requirements, including those for file naming. The PLOS ONE style templates can be found at

Answer:

We use the POS ONE Latex template (https://journals.plos.org/plosone/s/file?id=9a7a/plos_latex_template_v3.6.zip) and checked

that everything fulfils the style criteria. If this is not the case we would be glad if you could point us to some issues.

Answer:

Thank you for this comment. We shared our code using a static reference link (https://osf.io/8rkjb/?view_only=4352e24595d34cb29b763b71b018766d) and also uploaded the relevant documentation.

3. Please upload a copy of Supporting Information Figure/Table/etc. Supporting information S1 table, S2 table, S3 table and S4 table which you refer to in your text on pages 11 and 12.

Answer:

Because of the major changes to the paper the table numbers changed but we uploaded all referenced supplementary tables as separate files.

Answer:

We checked our references and we cite no retracted article. Unfortunately we found some minor errors in the references 3, 4, 9, 14 and 20.

These errors are fixed now.

5. While revising your submission, please upload your figure files to the Preflight Analysis and Conversion Engine (PACE) digital diagnostic tool, https://pacev2.apexcovantage.com/. PACE helps ensure that figures meet PLOS requirements. To use PACE, you must first register as user. Registration is free. Then, login and navigate to the UPLOAD tab, where you will find detailed instructions on how to use the tool. If you encounter any issues or have any questions when using PACE, please email PLOS at figures@plos.org. Please note that Supporting Information files do not need this step.

Answer:

We uploaded all figure files and they were converted to TIF.

Reviewer 1:

1. The authors have not clearly specified the background of the problem and main contributions of the work.

Answer:

In the introduction, we have now significantly extended the background and given examples of papers where we think our method would be useful.

2. The introduction section need to be properly organized with more clarity.

Answer:

We have added more details and tried to make it clearer what the paper is about. First, we motivate the usefulness of ML for data evaluation and its application in different fields. Next, we systematically list some problems and possible mistakes for data evaluation using ML. Finally, we propose our solution.

3. The experiment needs more regorios analysis. Moreover, hypertuning experiment as well as Ablation study can be included.

Answer:

We followed your advise and conducted an ablation study as well as using the confusion matrices to gain some insights on possible errors. We focused our interpretation of confusion matrices on differences between metrics because we aimed not to interpret the exemplary used SVM classifier, which can be chosen freely.

4. I suggest to add an additional experiment via error analysis that is usually helpful for model trst8ng with diverse conditions.

Answer:

We aimed to fulfil this request by above mentioned analysis of confusion matrices and the ablation study.

5. Have authors considered model evaluation via cross-domains? My suggestion is do more experimentation through cross-domain with small datasets.

Answer:

Our synthetic data does not belong to a specific domain, but we did evaluate other well-known real-world datasets from computer vision and medical research.

6. Finally, use a real-time case of both images and other datasets to measure efficacy of proposed method

Answer:

Unfortunately, the method proposed is not suitable for real time evaluation because it is only intended to find connections in data and to accurately specify how well an ML method would be able to predict new data if trained in the investigated dataset. The purpose is to be able to e.g. compare classifiers accurately or to prove the existence of weak connections in data which is for example for the field of education of great interest.

Reviewer 2:

1. Another widespread non-parametric technique is Random Forest. It would be interesting to mention this technique in the introduction chapter and highlight the reason for choosing the SVM algorithm.

Answer:

In the introduction, we have now explained why we have chosen the SVM for the demonstration of the method and briefly mentioned Random Forest (see lines 16 to 25 of the revised manuscript).

2. It is necessary to review punctuation, especially the use of commas. Revise long paragraphs as they tend to confuse the reader. I’ve pointed out in the comments on the digital file the passages where the writing should be revised.

Answer:

We attempted to use shorter paragraphs and revised punctuation.

3. When acronyms are presented for the first time in the text, their meaning should be presented.

Answer:

We revised the use of acronyms to better follow best practise.

4. Highlight the objectives of the study in chapter 1.

Answer:

Yes, we have included the study objectives in the introduction section.

5. Lines 62 to 85: a flowchart would illustrate the methodology used more effectively.

Answer:

We noticed that the explanation of synthetic data generation was unnecessarily complicated, so we revised it and added a flowchart (see Fig. 1).

6. Line 106: ”We do not examine dataset sizes below 25 data points because the performance is already not good for 25 data points”. Please detail this restriction.

Answer:

We have rewritten this part and explained our reasoning. A smaller size would not give better results and our results show that this dataset size is a good example of a dataset where the trained classifier may not generalise to more data points. 

Therefore, we do not expect further insights from a smaller dataset.

7. Line 153: Were the SVM parameters determined empirically? Please detail.

Answer:

No, we used the default parameters of the function in scikit-learn as a starting point and tried all possible values where applicable, or tried heavily modified values. The aim was to show the effect of hyperparameter tuning, not to achieve the best possible result. We added the details of how we obtained the parameters to revised version of the manuscript.

8. Line 188: ”However, the focus of this work is on the qualitative behavior and the demonstration of the evaluation methodology and, therefore, fewer permutations are used for reasons of computational time.” 

The parameter computational time appears repetitively in the text, making it clear that fewer permutations were chosen at the expense of less computational time. In this case, the use of cloud computing could solve this restriction.

Answer:

We rerun some calculations for the CESAR dataset so that now everything is calculated using 50 permutations. This recalculation as well as the evaluation of the MNIST and BCWD dataset took almost a month of computation time using a whole node on our cluster. The by far most time took the full rnCV because of the feature selection. It would be possible to use even more computing power to double or even triple the number of permutations, but we do not believe that this would improve our results, since we have been able to show that a higher number of permutations yields lower probabilities on average, but shows the same qualitative trend. It is therefore difficult for us to justify such an intensive use of resources.

9. Detail the applications used, as well as the hardware used.

Answer:

We have added a short paragraph at the end of the ’Evaluated data’ subsection about the hardware and applications used. 

10. The chapter reserved for discussions explores the results achieved. However, when compared with the results obtained by the authors cited in the references, the superior results achieved by the proposed methodology could be better substantiated.

Answer:

The introduction as well as discussion now contain text clarifying the merit of our proposed method as well as giving examples for use cases. 

11. Line 432: it would be interesting to create a chapter on conclusions.

What are your recommendations for future studies?

Answer:

We added a conclusion chapter and gave explicit recommendations.

Thank you again for taking the time to review our paper and for your valuable comments.

Yours sincerely,

Steffen Steinert

---

## [Editor Report · Decision Letter 1]

14 Mar 2024

A refined approach for evaluating small datasets via binary classification using machine learning

PONE-D-23-36197R1

Dear Dr. Steinert,

We’re pleased to inform you that your manuscript has been judged scientifically suitable for publication and will be formally accepted for publication once it meets all outstanding technical requirements.

Kind regards,

Sathishkumar Veerappampalayam Easwaramoorthy

Academic Editor

PLOS ONE
---

## [Editor Report · Acceptance letter]

30 Apr 2024

PONE-D-23-36197R1 

PLOS ONE

Dear Dr. Steinert, 

I'm pleased to inform you that your manuscript has been deemed suitable for publication in PLOS ONE. Congratulations! Your manuscript is now being handed over to our production team.

Kind regards, 

on behalf of

Dr. Sathishkumar Veerappampalayam Easwaramoorthy 

%CORR_ED_EDITOR_ROLE%

PLOS ONE